# mRNA Sequencing Reveals Upregulation of Glutathione S-Transferase Genes during *Acanthamoeba* Encystation

**DOI:** 10.3390/microorganisms11040992

**Published:** 2023-04-11

**Authors:** Alvaro de Obeso Fernández del Valle, Christian Quintus Scheckhuber, David Armando Chavaro-Pérez, Erandi Ortega-Barragán, Sutherland K. Maciver

**Affiliations:** 1Departamento de Bioingeniería, Escuela de Ingeniería y Ciencias, Tecnologico de Monterrey, Ave. Eugenio Garza Sada 2501, Monterrey 64849, Nuevo León, Mexico; 2Tecnologico de Monterrey, Escuela de Medicina y Ciencias de la Salud, Ave. Eugenio Garza Sada 2501, Monterrey 64849, Nuevo León, Mexico; 3Centre for Discovery Brain Sciences, Edinburgh Medical School, Biomedical Sciences, University of Edinburgh, Hugh Robson Building, George Square, Edinburgh EH8 9XD, Scotland, UK

**Keywords:** *Acanthamoeba*, glutathione S-transferase, encystation, cyst

## Abstract

Some members of the genus *Acanthamoeba* are facultative pathogens typically with a biphasic lifestyle: trophozoites and cysts. *Acanthamoeba* is capable of infecting the cornea, resulting in *Acanthamoeba* keratitis. The cyst is one of the key components for the persistence of infection. Gene expression during *Acanthamoeba* encystation showed an upregulation of glutathione S-transferase (GST) genes and other closely related proteins. mRNA sequencing showed GST, and five genes with similar sequences were upregulated after 24 h of inducing encystation. GST overexpression was verified with qPCR using the HPRT and the cyst-specific protein 21 genes as controls. The GST inhibitor ethacrynic acid was found to decrease cell viability by 70%. These results indicate a role of GST in successful encystation, possibly by maintaining redox balance. GST and associated processes could be targets for potential treatments alongside regular therapies to reduce relapses of *Acanthamoeba* infection.

## 1. Introduction

*Acanthamoeba* is well known to cause infections in the central nervous system [1,2] and the cornea [3,4] in humans. The infection of the cornea is called *Acanthamoeba* keratitis (AK), and it mainly infects contact lens wearers. There are no specific treatments for AK. Members of the genus *Acanthamoeba* usually have a biphasic lifestyle composed of a vegetative trophozoite and a latent cyst. The cysts are particularly problematic as they facilitate the persistence of the infection [3].

*Acanthamoeba* encystation consists of two stages; the first is characterised by autophagy and the degradation of proteins, and the second involves cyst-specific proteins that are translated, transforming the organism into a cyst [5]. Several factors have been studied during *Acanthamoeba* differentiation, including proteases [5,6], autophagy [7,8], cellulose synthesis [9], and cyst wall proteins [10,11].

The gene expression of encystation of different protozoa has been researched to better understand the underlying processes [12]. *Acanthamoeba* and *Entamoeba* encystation have been studied through several methods, including microarrays and RT-PCR [12,13,14,15]. Several authors have suggested that blocking encystation might be the key in dealing with encysting protozoan infections and their persistence [4,16,17,18].

In this paper, we identified glutathione S-transferases (GST) and related genes as factors vital for cyst viability and survival. In 1990, the first GSTs in *Acanthamoeba* were identified [19]. As with other organisms, the redox state, in which GSTs play an important role, is vital for life cycle changes in *Acanthamoeba* as it has been shown to be in several other organisms [19,20,21,22,23,24,25,26]. GSTs have already been studied as potential targets for other protozoan infections such as malaria [27]. 

## 2. Materials and Methods

### 2.1. Acanthamoeba Cell Culture

*Acanthamoeba castellanii* T4 strain 53 was used for most of the experiments. Strain 53 was isolated in the laboratory from soil samples collected from Silverburn, Scotland (55°49′57.5″ N 3°15′41.6″ W). AX2 media was used for *Acanthamoeba* axenic cultures. It consisted of Bacto tryptone (14.3 g/L), yeast extract (7.15 g/L), glucose (15.4 g/L), Na_2_HPO_4_ (0.51 g/L), and KH_2_PO_4_ (0.486 g/L), with a pH of 6.5.

### 2.2. Encystation and RNA Extraction

To induce encystation, cultures were washed with PBS and Neff’s encystation media (NEM) was added. NEM consisted of 0.1 M KCl, 8 mM MgSO_4_, 0.4 mM CaCl_2_, 1 mM NaHCO_3_, and 20 mM 2-amino-2-methyl-1,3-propanediol, pH 8.8. I. 2-amino-2-methyl-1,3-propanediol could be substituted with 10 mM Tris-HCl pH 8.0 with similar results [28]. Cells were collected at 4 different timepoints: 0, 24, 48, and 72 h after exposure to NEM. Once each timepoint was reached, cells were collected, centrifuged (150 g for 10 min), and washed with Neff’s saline. RNA extraction was performed with QIAGEN RNeasy isolation kit. RNA quality was tested by observing agarose gels and measuring their purity with QUBIT RNA BR (Broad-Range) Assay Kit (Thermo-Fisher Scientific, Loughborough, UK). All of the experiments were performed in triplicates.

### 2.3. mRNA Sequencing

Libraries were prepared for an automated TruSeq stranded mRNA-seq from total RNA. The sequencing data generation was made with HiSeq-4000 75PE. These experiments were carried out by Edinburgh Genomics, The University of Edinburgh. The reference genome (FASTA and GTF files) from *A. castellanii* was obtained from ENSEMBL Protists [29]. The genome was indexed, and the reads aligned using STAR 2.5.3a software [30]. 

### 2.4. Differential Expression Analysis

The differential expression analysis was performed using R studio and edgeR [31]. Counts per gene were generated using featureCounts software with reverse stranded reads [32]. The data was normalised using the trimmed median of M values or TMM [33]. Data was filtered by applying a counts per million (CPM) threshold of 0.1. The dispersion was estimated assuming that biological coefficient variation is constant and fitted to generalised linear models using the quasi-likelihood approach.

The differential expression analysis was performed using Limma’s “makeContrasts” and “topTags” functions [34]. Differentially expressed genes were found using criteria of a Log2 fold change (Log2FC) value over 2 and under -2 with an FDR threshold of 0.05. 

Later, to select overexpressed genes of interest, we focused on Log2FC genes that had values over 5. Once selected, hypothetical proteins were searched using BLAST. Cyst-specific protein 21 (CSP21) was used as a positive control for all encystation profile experiments [10]. 

### 2.5. qPCR Expression Analysis

Sequencing results were verified using qPCR. Primers were developed for GST-identified genes using HPRT and CSP21 as a control [35]. Primer design was performed with Primer-BLAST and Primer3Plus [36,37]. Primers focused on the GST gene reported in the literature (ACA1_116240). Parameters were optimized for Tm between 57 and 59 °C, and a length of 75 to 200 bp. Table 1 shows the sequences and characteristics of the primers used. qPCR was performed using SYBR Green Universal Master Mix from Applied Biosystems following the manufacturer’s instructions. Cycling conditions included 95 °C for 10 min, followed by 45 cycles of 95 °C for 10 s, 60 °C for 15 s, and 72 °C for 20 s.

Data were analyzed using the 2−ΔΔCT method [38]. HPRT was used as a calibration gene, and CSP21 was used as a positive control since it increases expression during encystment [10,35]. 

### 2.6. Glutathione S-Transferase (GST) Inhibitors

*Acanthamoeba* cultures were grown to confluence. Media were discarded, and cultures were washed with Neff’s saline. Cultures grown in AX2 media were used as a negative control, while positive controls were obtained by adding NEM to cultures. The treatments were created with cultures grown in NEM supplemented with GST inhibitors: etacrynic acid (25 µM and 250 µM concentrations) and sulfasalazine (100 µM and 1 mM). Three cultures for each treatment were maintained at room temperature for 72 h. Trophozoite and cysts were counted in each culture using a haemocytometer. Cyst viability was tested using the trypan blue exclusion method [39]. 

## 3. Results

mRNA sequencing for encystment in *Acanthamoeba* was performed to identify genetic factors involved in the process. In total, 13,271 transcripts were analyzed by comparing the vegetative stage to encystment. Using a Log2FC cut-off of 1, 2026 transcripts were identified as downregulated during encystment after 24 h, while 1557 were identified as upregulated. As a control, the expression profile of the CSP21 was studied. CSP21 was upregulated after 24 h with a Log2FC over 5. Transcripts with a similar expression profile with a Log2FC over 5 were researched. In total, 56 upregulated transcripts were identified with such characteristics. The selected genes were queried in AmoebaDB. Of the 56 genes, 40 were registered as hypothetical proteins. BLAST analysis was performed with these hypothetical genes, comparing the values of identity from the genomic sequence, predicted mRNA, and predicted protein. From the BLAST results, five protein sequences had identities with a value over 50% compared to the known “*Acanthamoeba* glutathione S-transferase, C-terminal domain containing protein” (ACA1_116240). The Log2FC values of the genes after 24, 48, and 72 can be seen in Table 2. The data are compared to the values obtained from the original GST C-terminal domain containing protein and CSP21. The identity values obtained with BLAST can be seen in Table 3. The five genes are recorded with their gene IDs from AmoebaDB: ACA1_ 188370, ACA1_247090, ACA1_096640, ACA1_022350, and ACA1_374130.

After mRNA sequencing, the results were verified using qPCR. The experiments confirmed the overexpression of GST-related genes after 24 h of inducing encystment. The results of qPCR can be observed in Figure 1.

The qPCR results confirmed the RNAseq differential expression profile. GST showed an overexpression fold change of 2.71 after 24 h compared to the control belonging to the trophozoite stage. The fold change for CSP21 was 6.86. 

Once genes related to GST were identified, cultures were treated with GST inhibitors. Trophozoites and encysting cultures were treated with GST inhibitors and incubated for 72 h. Trophozoites did not show any difference in viability after 72 h of incubation with the inhibitors (data not shown). During encystment, a significant decrease in cell viability occurred when treating the cultures with GST inhibitors. Both inhibitors reduced cell viability. Ethacrynic acid showed the largest effect as it lowered viability by 70% of the cells, while sulfasalazine prevented cyst viability in slightly over 40% of the cysts. The results can be seen in Figure 2.

## 4. Discussion

GSTs belong to a protein family that normally plays a role in normal cell metabolism and detoxification by maintaining redox balance. Redox balance and fluctuations have been identified as important factors in the life cycle of several organisms such as yeast, plants, and mammalian cells [20,23]. Of note is the pronounced up-regulation of genes that encode for GST or related proteins in the present study. Biochemically speaking, these enzymes have various functions in the detoxification of xenobiotics and defense against certain secondary reactive oxygen species (ROS) and lipid peroxidation products and are also able to bind and store a variety of compounds such as fatty acids in a non-enzymatic ‘ligandin’ function [40,41,42,43,44]. However, given their function in maintaining the redox balance in the cell as mentioned before, GST can also exert a documented pro-oxidative function by depleting the redox capacity of the GSH/GSSG pool, with clear consequences for the structural properties of the mitochondrial population in axons [45]. In the present study, no genes encoding antioxidant proteins were found to be up-regulated, and in some cases were even down-regulated (alternative oxidase, AOX, results not shown); thus, a pro-oxidant effect by GSTs might drive *Acanthamoeba* encystment by promoting the fragmentation of mitochondria. Punctate mitochondria are necessary for autophagy to be effective in recycling them as they are more readily engulfed by phagophore membranes [46,47,48]. Autophagy induction is a cellular hallmark during encystment of amoebae [5,8,49,50,51]. Other microorganisms also display transitions of mitochondrial morphology during certain developmental processes. During the sporulation of *Saccharomyces cerevisiae,* pronounced mitochondrial fragmentation takes place [52]. Furthermore, in the filamentous fungus *Podospora anserina,* it was experimentally demonstrated that mitochondrial fission is necessary for allowing ascospores to germinate efficiently [53]. In general, the processes of encystment and sporulation might have several biological principles in common.

Previously, Lloyd speculated that preserving redox balance is vital to maintain cell viability in *Acanthamoeba*, and the evidence obtained from inhibiting GSTs during encystment supports this [50]. Encystment in *Acanthamoeba* begins with the degradation of proteins and autolysis, such as the partial breakdown of actin in the beginning of the process, producing several components that need to be eliminated for the continual viability of the cyst [5]. The inhibition of GST alters the redox balance necessary for viable cysts. In other organisms, oxidative stress and antioxidants can induce the transcription of GST genes, providing protection from environmental and chemical factors [54]. Therefore, the anti-oxidative effects of GST upregulation during the encystment process cannot be ruled out. 

We suggest that GSTs might be interesting targets for the treatment of *Acanthamoeba*-mediated infections. As a proof-of-principle, GST inhibitors have been successfully employed as antiparasitic agents, e.g., for inactivating the malarial parasite *Plasmodium falciparum* [27] and disrupting the larval stages of the porcine nodule worm *Oesophagostomum dentatum* [55]. Furthermore, GSTs in protozoans have been linked to drug resistance [56]. Additionally, thiols and enzymes of redox metabolism, antioxidant enzymes, and encystment pathways have been suggested as potential drug targets for *Entamoeba histolytica*, *Acanthamoeba polyphaga,* and *Naegleria fowlerii* [22]. Encystment processes and certain aspects of drug resistance in different protozoans might have a common evolutionary ancestor as both are survival mechanisms with similar molecular mechanisms.

Ethacrynic acid has been tested for toxicity and potential use in the eye as it has been shown to increase the facility of outflow [57]. From this, ethacrynic acid was suggested as an anti-glaucoma drug and even underwent pre-clinical trials [58,59]. There are some adverse effects to topical application after prolonged exposure and alternatives have been researched [60]. However, more studies are required in regard to AK since applications of ethacrynic acid were stopped for glaucoma after one clinical trial failed to reach the desired outcomes [61]. One of the problems regarding glaucoma is that drugs have to be long lasting, which is not necessarily the case for AK infection. Sulfasalazine has been tried as a potential or actual treatment in different ocular diseases such as anterior uveitis [62,63], ocular cicatricial pemphigoid [64], and posterior capsule opacification [65].

Although GST inhibitors could be used as treatment, they can be potentially difficult to establish due to the fact that humans as well as other eukaryotes produce GSTs. As with many other therapeutic targets, the challenge is targeting the pathogen but not the host [66]. Moreover, enzyme inhibitors never reach 100% efficiency [67]. In this case, ethacrynic acid was capable of reducing viability by 70%. Developing siRNA to target the specific GSTs exclusive from *Acanthamoeba* might be an option.

Additionally, if the aforementioned link between mitochondrial fragmentation and *Acanthamoeba* encystment is demonstrated experimentally, the use of inhibitors of the division process similar to mitochondrial-division inhibitor 1 (Mdivi-1) [68] could be attractive for testing as treatments for keratitis. Additionally, as GSTs are related to sulphur metabolism, studies regarding sulphur metabolism, detoxification, and encystment might be needed as the oxidative detoxification of hydrogen sulphide by *A. castellanii* has been reported [69].

To conclude, cysts are the main reason for infection persistence in AK. Therefore, using GST inhibitors alongside other treatments might provide a synergistic treatment. We have shown that GSTs play an important role in the encystment process. Here, we propose inhibiting the effect of GSTs alongside regular therapies against AK to help reduce the number of relapses of the disease.

## Figures and Tables

**Figure 1 microorganisms-11-00992-f001:**
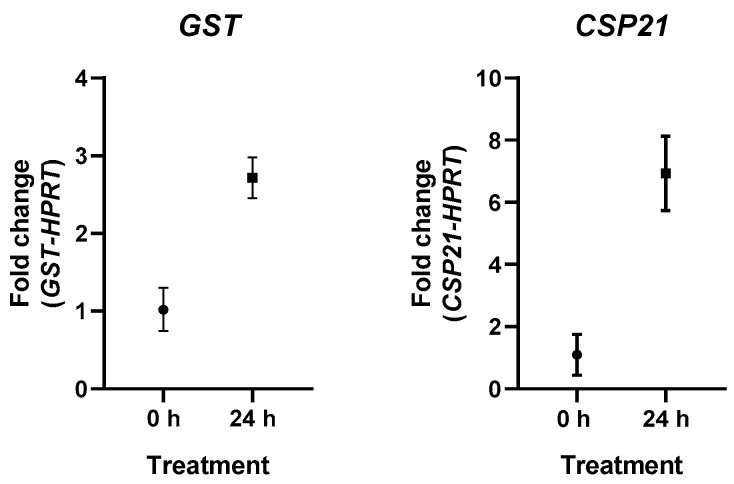
Differential expression of GST (**left**) and CSP21 (**right**) after 24 h of encystment stimulus. Each graph compares treatments at 0- and 24-h, representing fold change in expression; error bars represent standard error of the mean.

**Figure 2 microorganisms-11-00992-f002:**
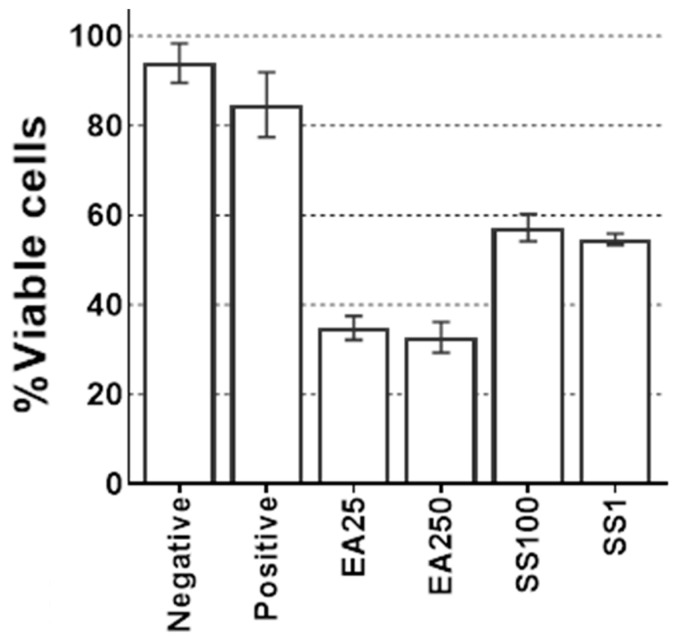
Cell viability of cultures encysting with GST inhibitors. Values represent percentage of cell viability. Negative control was used as a regular culture with AX2. The positive control is the encysting culture with NEM. EA25: Ethacrynic acid 25 µM. EA250: Ethacrynic acid 250 µM. SS100: Sulfosalazine 100 µM. SS1: Sulfosalazine 1 mM. Error bars represent the standard error of the mean.

**Table 1 microorganisms-11-00992-t001:** qPCR primers and their characteristics.

Gene	Sequence (5′-3′)	Tm	Amplicon Lenght	Source or Accesion Number
GST	F: CAAGTGCTACCCCAAGGAC	57.75	162 bp	NW_004457554
	R: CCCTTCTCGTCCGGGTAG	58.48		
CSP21	F: ACTTTGGCGACAAGGTGTG	58.6	80 bp	XM_004337011
	R: CGACACGTCGTCCCTCT	58.31		
HPRT	F: GGAGCGGATCGTTCTCTG	58.4	201 bp	[35]
	R: ATCTTGGCGTCGACGTGC	58.4		

**Table 2 microorganisms-11-00992-t002:** Log2FC values for hypothetical proteins related to GST c-terminal compared against gene expression at 0 h. The table includes values for the original glutathione S-transferase C-terminal domain containing protein and the cyst-specific protein 21 as reference.

Gene_ID	Description	24 h	48 h	72 h
ACA1_116240	GST C-terminal domain containing protein	4.7154	0.4877	0.4623
ACA1_075240	Cyst-specific protein 21	6.5435	3.5723	1.7860
ACA1_022350	Hypothetical protein	8.4458	3.9417	1.4994
ACA1_096640	Hypothetical protein	7.0754	4.2638	1.8371
ACA1_188370	Hypothetical protein	10.0624	6.1620	3.4728
ACA1_247090	Hypothetical protein	7.5066	2.7215	0.8404
ACA1_374130	Hypothetical protein	6.8663	2.5504	0.2569

**Table 3 microorganisms-11-00992-t003:** Identity values for the five hypothetical proteins in relation to glutathione S-transferase c-terminal domain. The values obtained from the predicted protein, the predicted RNA, and the genomic sequence as obtained from AmoebaDB.

Gene_ID	Predicted Protein	Predicted RNA	Genomic
ACA1_188370	74%	73%	75%
ACA1_022350	NA	75%	75%
ACA1_247090	54%	72%	68%
ACA1_096640	54%	74%	73%
ACA1_374130	79%	81%	79%

## Data Availability

mRNA data are available upon request.

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
