# Peer review of "mRNA Sequencing Reveals Upregulation of Glutathione S-Transferase Genes during *Acanthamoeba* Encystation"

_microorganisms, 2023, doi:10.3390/microorganisms11040992_

Round 1

Reviewer 1 Report

The manuscript entitled "mRNA sequencing reveals upregulation of glutathione S-transferase genes during Acanthamoeba encystation" is a well written manuscript. It describes the role played by glutathione S-transferase (GST) in the encystment of A. castellanii trophozoites. When researchers placed trophozoites in encystment conditions, they observed the overexpression of five related genes.They also test for the presence of ethacrynic acid and sulfazalacine GST inhibitor, and observed the decrease in trophozoite viability by 70% and 40% respectively. Finally, the authors propose using GST inhibitors as a possible treatment against this protozoan.

Questions

1. Would you think that incubating A. castellanii trophozoites with GST inhibitors would decrease the expression of genes related to this enzyme?

2. I think it would be necessary to test the use of GST inhibitors in in vitro studies using cell lines as well as in animal models, since they also having GST and could cause cell damage.

Author Response

We do appreciate the comments from the reviewer.

Would you think that incubating A. castellanii trophozoites with GST inhibitors would decrease the expression of genes related to this enzyme?

This is an excellent question. We speculate that the expression of the genes encoding these enzymes would increase to compensate for their inhibition. Alternatively, there might be no specific response as these genes are not highly expressed. We will look into this in further studies (see below).

 I think it would be necessary to test the use of GST inhibitors in in vitro studies using cell lines as well as in animal models, since they also having GST and could cause cell damage.

Yes, we agree, but these experiments are beyond the scope of our present contribution. We do consider this line of research very appropriate, so we expanded the discussion section ( e.g., use in the eye as treatment for different ocular diseases) accordingly. Furthermore, we are considering to investigate the cytotoxicity of these inhibitors in corneal epithelial cells in future works. For these we would subject the cell lines to conditions of oxidative stress through the application of hydrogen peroxide (H2O2) and/or tert-butyl-hydroperoxide, to test for potentiated cell injury in the presence and absence of the GST inhibitors. This research could be extended to include animal models.

Reviewer 2 Report

In the case of fig 2. labeling is missing and a statistical test can be added for better understanding.

For sequencing how many replicates are used for each time point? It is not clear.

Tables 2 and 3 are wrongly referenced in the text.

In the case of table 2 (according to the paper), the log2FC value for 0h should be added to compare the data

Grammar mistakes and typo errors are present.

Please thoroughly review the whole paper.

Author Response

We appreciate the comments by the reviewer. There were several mistakes that have now been corrected. Most of the suggested corrections have been implemented.

Regarding the tables, we corrected the errors on the labels. We also added the number of replicates. To figure 2, we have added the missing label. We did not add any other statistical analysis as a t test would only report differences and we believe that the SEM shows that. To table 2 we added in the header that all data are compared to 0 hours which should clarify the point. The whole paper was proofread to check for grammar mistakes by a native speaker of English, Dr. Sutherland Maciver.